# Blind Source Separation of Intermittent Frequency Hopping Sources over LOS and NLOS Channels [note 1]

**DOI:** 10.3390/e25091292

**Published:** 2023-09-03

**Authors:** Anushreya Ghosh, Annan Dong, Alexander Haimovich, Osvaldo Simeone, Jason Dabin

**Affiliations:** 1CWiP, New Jersey Institute of Technology, Newark, NJ 07102, USA; ad372@njit.edu (A.D.); haimovic@njit.edu (A.H.); 2KCLIP Lab., Department of Engineering, King’s College London, London WC2R 2LS, UK; osvaldo.simeone@kcl.ac.uk; 3Naval Information Warfare Center Pacific, San Diego, CA 92152, USA; jason.a.dabin.civ@us.navy.mil

**Keywords:** blind source separation, frequency hopping, direction of arrival, 3GPP spatial channel model, hidden markov model

## Abstract

This paper studies blind source separation (BSS) for frequency hopping (FH) sources. These radio frequency (RF) signals are observed by a uniform linear array (ULA) over (i) line-of-sight (LOS), (ii) single-cluster, and (iii) multiple-cluster Spatial Channel Model (SCM) settings. The sources are stationary, spatially sparse, and their activity is intermittent and assumed to follow a hidden Markov model (HMM). BSS is achieved by leveraging direction of arrival (DOA) information through an FH estimation stage, a DOA estimation stage, and a pairing stage with the latter associating FH patterns with physical sources via their estimated DOAs. Current methods in the literature do not perform the association of multiple frequency hops to the sources they are transmitted from. We bridge this gap by pairing the FH estimates with DOA estimates and labeling signals to their sources, irrespective of their hopped frequencies. A state filtering technique, referred to as hidden state filtering (HSF), is developed to refine DOA estimates for sources that follow a HMM. Numerical results demonstrate that the proposed approach is capable of separating multiple intermittent FH sources.

## 1. Introduction

Frequency hopping (FH) spread spectrum signals have been widely studied and adopted for wireless communications due to a multitude of advantages, such as their low probability of detection and their inherent robustness to jamming [1,2,3]. Estimating and tracking parameters of multiple FH signals have important applications in both civilian and military fields, such as collision avoidance [4,5], cognitive radio [6,7], and interception of non-cooperative communications [8,9]. The estimation of FH signal parameters for the purpose of intercepting non-cooperative sources is the focus of this work.

The FH sources assumed in this paper transmit at frequencies that change pseudo-randomly within a block of spectrum. Parameters such as hop time, hopping pattern, and frequencies are random and unknown at the receiver. The localization and separation of multiple FH sources without knowledge of these parameters is posed as a blind source separation (BSS) problem [10].

To perform BSS of multiple FH sources, it is not sufficient to produce a frequency versus time map of power of the observed signals; it is also necessary to associate the signals to physical sources. We refer to the task of associating frequency hops to a source as the problem of labeling of FH signals. Since frequency hop estimates cannot determine which of several sources transmitted the signal, it is necessary to estimate information that is source specific and extraneous to the FH pattern. To achieve this, one can leverage the knowledge that all signals transmitted by a source via line-of-sight (LOS) propagation have the same direction of arrival (DOA) information.

Estimating the DOAs of the sources is made more challenging if signals are received via multipath propagation. Measurement data analysis in [11] demonstrates that physical structures in the channel act as secondary sources, forming separable clusters with narrow angular spread around the clusters [12,13,14]. Several cluster-based channel models can be found in the literature, such as the 3GPP Spatial Channel Model (SCM) [15,16,17], WINNER II [18], and the 3GPP Clustered Delay Line (CDL) model [19]. In [15,16,17], the SCM is defined for different scenarios, namely suburban macro, urban macro and urban micro. The urban micro channel propagation environment deals with LOS sources, and the other two propagation environments take the effects of multipath propagation into account and deal with non-line-of-sight (NLOS) sources. In this work, signals are observed over channels that follow the SCM model for three propagation environments: (i) LOS, (ii) single-cluster, and (iii) multiple-cluster settings.

### 1.1. Related Work

Among various approaches used to solve the BSS problem of FH sources is time-frequency analysis (TFA) [20,21,22,23]. TFA methods are applied to study representations of the received FH signal in both time and frequency domains. However, as captured by the uncertainty principle, it is not possible to reach good time and frequency resolutions simultaneously [24]. TFA methods also suffer from cross-term interference and spectral leakage, resulting in high SNR requirements [25].

TFA-based methods have been used as exploratory tools towards more refined solutions to blind estimation of hop timings and frequencies. When only one FH signal is present, reference [26] proposes to first apply TFA to estimate the hopping pattern, and, subsequently, a particle filter operates on the initial estimation. The initial estimation of hopping patterns in [26] depends on TFA-based methods, and therefore has high signal-to-noise (SNR) requirements. A blind maximum-likelihood (ML)-based iterative algorithm is proposed in [27] that estimates hop timing and frequency hops for a single user. The ML-based algorithm has been shown to have lower SNR requirements than TFA-based approaches in [27]. However, the formulation in [27] cannot be generalized to multiple FH signals.

For multiple FH signals, the method proposed in [28] implements a dynamic programming-based ML estimator that yields estimates of joint hop timings and frequencies. An approach based on sparse linear regression is introduced to estimate the hop timings and frequencies of multiple FH signals in [29]. Each hop in [28,29] is treated as a distinct source. This method is not suitable for grouping frequency hops according to physical sources. To associate frequency hops to a source and label FH signals, the DOAs of the sources are estimated as they are source specific extraneous to the FH pattern.

In reference [30], a two-step approach is introduced to estimate DOA, hop timings, and frequency hops for multiple sources. A TFA method is applied to signals received by a uniform linear array (ULA) to identify a hop-free duration for DOA estimation. After the DOAs are recovered, joint estimation of hop timings and frequencies is performed for each signal originating from the same DOA. However, as mentioned earlier, references like [30] that rely on TFA-based approaches for initial estimations suffer from cross-term interference and high SNR requirements. The joint estimation of FH parameters and DOA for multiple sources is studied in [31,32,33,34] under the assumption that all sources are active throughout the entire observation interval. Additionally, in reference [34], the hop periods are assumed uniform. None of the approaches in [31,32,33,34] are able to incorporate sources that are sparse spatially and have intermittent activity, and probabilistic source models, such as Markov models. A comprehensive summary of the approaches in the current literature and their limitations can be found in Table 1.

### 1.2. Main Contributions

This work addresses the problem of BSS of FH sources, which are stationary, spatially sparse, have activity that is intermittent and follow a hidden Markov model. Three different propagation environments of the SCM channel models are considered, namely (i) LOS, (ii) single-cluster, and (iii) multiple-cluster settings.

Blind source separation of frequency hopping sources is a problem that has been tackled in RF communications before. However, it is not sufficient to just produce a frequency versus time of observed signals; it is also necessary to associate the signals to physical sources. Current methods lack the association of multiple frequency hops to the sources they are transmitted from. We fulfill this gap in our paper by pairing the FH estimates with DOA estimates and labeling signals to their sources, irrespective of their hopped frequencies. This is performed for both LOS and NLOS channels.

Compared to the conference version [35], this paper studies sources that transmit FH signals over both LOS and NLOS channels; and, compared to [36], this paper includes a comparative study of two filtering approaches that refine the estimations to provide improved source labeling.

The main contributions of this paper are as follows:1.A sparse representation framework is introduced to determine frequency hops and DOA of propagation paths of signals emitted by physical RF sources. FH and DOA estimations are posed and solved as sparse representation problems;2.A method is developed to associate FH and source activity of multiple sources, thus effectively achieving blind source separation;3.It is shown that applying hidden state filtering (HSF) improves BSS performance;4.An algorithm is developed that combines HSF with the estimation of HMM parameters, implementing the filtering without prior knowledge of the model parameters.

The rest of the paper is organized as follows. The system model with aperiodic FH and the hidden Markov source model are presented in Section 2. In Section 3, we propose the approach to separate FH, source activity and assign labels to signals transmitted from different sources. In Section 4, simulation-based numerical results demonstrate the performance of the proposed approach, and Section 5 reports our conclusions.

Notation: Notation 1:T denotes the sequence 1,2,…,T. Vectors are denoted by boldface lower case letters, such as x. All vectors are assumed to be column vectors. Matrices are denoted by boldface upper-case letters, such as X. The transpose of X is denoted as $X^{\prime }$.

## 2. System Model

### 2.1. Setting and Channel Model

We consider *N* sources capable of transmitting intermittent FH signals. These sources switch carrier frequency in a randomized fashion across multiple frequency hops. Signals emitted by the sources are received at a uniform linear array (ULA) with *J* receiving antenna sensors spaced at uniform *d* intervals.

The total number of sources *N* may be larger than the number of sensors *J*, but the number of active sources at any given time is lower than the number of sensors. The observation interval is *T* discrete time units, and, without loss of generality, the sampling interval corresponds to one time unit. The cumulative time a source is active is a small fraction of the observation interval. Source activity is assumed to be governed by a Markov model, as described below. The on–off patterns of each source change slowly, making transitions intermittent, but smooth. The intermittent activity of the sources is illustrated in Figure 1.

A hop duration is the period of time between two consecutive switches of the carrier frequency of a source. The time of the *k*-th frequency switch is denoted $t_{k}$, and thus the duration of the corresponding hop is $\left(t_{k+1}-t_{k}\right)$. When multiple sources are active, the hop duration is determined by the source with the shortest time between frequency hops. An example is shown in Figure 2.

Sources are assumed to be observed over channels that follow the SCM. The SCM is a statistical channel model developed by 3GPP for three propagation environments: (i) LOS, (ii) single-cluster, and (iii) multiple-cluster settings [15,16,17]. Over an LOS propagation environment, a source is observed with a single DOA. With the single-cluster environment, a source is observed over a cluster of possibly unresolvable multipath propagation paths. For the multiple-cluster environment, the multipath results in multiple clusters of propagation paths. In this work, BSS of FH sources is considered over each of the three propagation environments.

For the SCM with multiple clusters, assuming synchronized sensors, the signal received at the *j*th sensor in the time interval $t_{k}\leq t<t_{k+1}$ corresponding to the *k*th hop is given by
(1)$$y\left(j,t\right)=\sum_{i=1}^{N_{k}}\left\{\sum_{l=1}^{L_{i}}\sqrt{\frac{P_{l}}{M}}\sum_{m=1}^{M}c_{i,l}^{\left(m\right)}\left(t\right)a^{j}\left(\theta_{i,l}^{\left(m\right)}\right)\right\}h^{t}\left(f_{i}\right)+w\left(j,t\right),$$
where $N_{k}$ is the number of active sources during the *k*th hop ($N_{k}\ll N$); $L_{i}$ is the number of clusters for the *i*th source; $P_{l}$ is the power of the *l*th cluster of the *i*th source which is normalized so that the total average power for all clusters is equal to one; *M* is the number of unresolvable multipaths per cluster that have similar characteristics. The number of sources that could be active during the *k*th hop is less than the number of sensors *J*, i.e., $N_{k}<J$. The variable $h(f_{i})$ is the frequency mode of the *l*th cluster of the *i*th source, and $a(\theta_{i,l}^{\left(m\right)})$ is the spatial mode of the *m*th multipath of the *l*th cluster of the *i*th source. $a^{j}\left(\cdot \right)$ and $h^{t}\left(\cdot \right)$ denote the corresponding *j*th and *t*th powers; $c_{i,l}^{\left(m\right)}\left(t\right)$ is the complex amplitude of the *m*th multipath of the *l*th cluster of the *i*th source; and w(m,t) is the additive zero-mean complex Gaussian noise with variance $\sigma_{w}^{2}$. The frequency mode is given by $h\left(f_{i}\right)=e^{j2\pi f_{i}}$, with $f_{i}\in \left[f_{\min},f_{\max}\right]$ being the carrier frequency of the *i*th source during the *k*th hop. The hop frequencies are measured relative to the carrier frequency of the receiver. The spatial mode is given by $a\left(\theta_{i,l}^{\left(m\right)}\right)=e^{j2\pi \left(d/\lambda \right)\sin\left(\theta_{i,l}^{\left(m\right)}\right)}$, where d/λ is the spacing between antennae expressed in units of wavelength λ, and $\theta_{i,l}^{\left(m\right)}$ is the DOA of the *m*th multipath of the *l*th cluster of the *i*th source. The DOA $\theta_{i,l}^{\left(m\right)}$ can be decomposed as $\theta_{i,l}^{\left(m\right)}=\theta_{i,l}+\vartheta_{i,l}^{\left(m\right)}$, where $\theta_{i,l}$ is the mean DOA of the *l*th cluster of the *i*th source, and $\vartheta_{i,l}^{\left(m\right)}$ is the deviation from the mean DOA, which is modeled as an i.i.d. Gaussian random variable with zero mean and variance $\sigma_{\theta }^{2}$. For the single-cluster propagation environment, $L_{i}=1$ and $\vartheta_{i,l}^{\left(m\right)}\neq 0$, whereas for the LOS propagation environment, $L_{i}=1$ and $\vartheta_{i,l}^{\left(m\right)}=0$.

For hop *k* and the corresponding time interval, $t_{k}\leq t<t_{k+1}$, observations y(j,t) are collected in a $J\times (t_{k+1}-t_{k})$ matrix $Y_{k}$. Combining observations across all hops, matrices $Y_{k}$ are concatenated to form the J×T observation matrix Y.

### 2.2. Source Activity Model

Sources are subject to intermittent activity. Activity is also “smooth”, i.e., the on–off patterns of sources vary slowly. A source’s activity pattern is assumed to be governed by a hidden Markov model (HMM). The HMM has been used in many instances to model communication systems [37,38,39,40]. An HMM imprints a memory on the system that is used to mimic the behavior of a communication system’s channel coding.

The activity pattern of a source is represented by a binary state sequence s(t) which indicates whether at time *t* a source is active (s(t)=1) or not (s(t)=0). As discussed previously, a physical source may be observed over multiple DOAs. The term “source activity pattern” is used here generically, and it refers either to a physical source or an individual DOA. It is assumed that the individual DOA activity patterns associated with a physical source are “similar”. Two activity patterns associated with different DOAs are considered “similar” if they match over a prescribed fraction of the time samples at which their values are 1.

A diagram representing the HMM of a source is shown in Figure 3. In the figure, the sequence s(1:T) represents the hidden states, while sequence z(1:t) represents the observed sequence. Like state symbols, observation symbols are binary, z(t)∈{0,1}, where z(t)=1 indicates the observed activity of the source at time *t*, and z(t)=0 indicates that source was inactive at time *t*. In addition to states and observations, an HMM is characterized by state transition probabilities, observation symbol probabilities, and initial state probabilities [41]. These are specified below for a source.

State transition probabilities are represented by the state transition matrix $A=\{a_{ij}\}$, j=0,1 where the state transition probability distribution is given by $a_{ij}=P\left(s\left(t\right)=j|s\left(t-1\right)=i\right)$. Observation symbols probabilities are represented by the observation matrix $B=\{b_{j}\left(k\right)\},j,k=0,1$ where $b_{j}\left(k\right)$ denotes the observation symbol probability distribution in state *j*, $b_{j}\left(k\right)=P\left(z\left(t\right)=k|s\left(t\right)=j\right)$. We use notation $b_{j}\left(z(t)\right)$ to indicate the probability of observed value z(t) conditioned on s(t)=j, $b_{j}\left(z(t)\right)=P\left(z\left(t\right)|s\left(t\right)=j\right)$. The initial state probability distribution is $\pi =\{\pi_{0},\pi_{1}\}$, where $\pi_{0}=P\left(s\left(1\right)=0\right)$ and $\pi_{1}=P\left(s\left(1\right)=1\right)$. The parameters of an HMM are succinctly denoted by Ω, where Ω=(A,B,π).

## 3. Method

The signal model in (Equation 1) has three sets of unknowns, namely DOAs θ, frequencies *f*, and complex amplitudes *c*. Each physical source may emit multiple frequencies as part of an FH pattern, and it may be observed over multiple DOAs for multipath channels. The goal of the process presented in this section is to determine physical sources, and for each physical source to determine an activity pattern and a FH pattern. Two main ideas are that even though a physical source may be observed over many DOAs, the individual DOA activity patterns of a physical source are “similar”. DOA information is paired with FH information to associate FH patterns with physical sources. In the following, an approach is proposed that uses the received signal matrix Y to estimate FH and activity patterns over the course of the observation interval *T*. This approach includes an FH estimation stage, a DOA estimation stage, hidden state filtering to refine DOA estimates, and a pairing stage that combines information from the previous stages to associated FH patterns with physical sources. Figure 4 elucidates the relations between the signal processing tasks developed in this section. Each of the processing tasks is detailed next.

### 3.1. FH Estimation

The estimation of the frequency hops at each time instant 0≤t≤T is posed as a sparse representation problem. In the following, we define a dictionary matrix, a matrix of unknowns, a measurement matrix, and a noise matrix. For setting up the dictionary, we let set $F=\{f_{1},f_{2},\ldots ,f_{G_{f}}\}$, with cardinality $G_{f}\gg N_{k}$, comprise all possible hop frequencies $f_{i}$. The FH estimation stage samples the frequency by using this grid of frequencies *F*. The frequency grid is used to define the $G_{f}$-length modal vector ${\tilde{h}}_{f}\left(t\right)$ at sampling instant *t*:(2)$${\tilde{h}}_{f}\left(t\right)=\left[h^{t}\left(f_{1}\right),h^{t}\left(f_{2}\right),\ldots ,h^{t}\left(f_{G_{f}}\right)\right].$$

Next, this vector is expanded to include all *T* sampling instants in a $TG_{f}$-length vector $h_{f}\left(t\right)={\underset{\;}{[}\underset{\underset{t}{︸}}{0_{G_{f}}^{\prime },\ldots ,0_{G_{f}}^{\prime }},{\tilde{h}}_{f}^{\prime }\left(t\right),\underset{\underset{T-t-1}{︸}}{0_{G_{f}}^{\prime },\ldots ,0_{G_{f}}}\underset{\;}{]}}^{\prime }$, where $0_{G_{f}}$ is a vector containing $G_{f}$ zeros. The FH modal dictionary $H_{f}$ is then defined at the $T\times TG_{f}$ matrix
(3)$$H_{f}={\left[h_{f}\left(1\right),h_{f}\left(2\right),\ldots ,h_{f}\left(T\right)\right]}^{\prime }.$$

The selection of matrix $H_{f}$ is determined by the choice of frequency grid *F* via the relation between ${\tilde{h}}_{f}\left(t\right)$, $h_{f}\left(t\right)$ and $H_{f}$. We let $x_{f}\left(j,t\right)$ be the $G_{f}$-length vector of unknown complex amplitudes associated with the grid frequencies at time *t* and sensor *j*. This vector is then expanded to form the $TG_{f}$-length vector that includes all *T* time instants $x_{f}^{*}\left(j\right)={\left[x_{f}^{\prime }\left(j,1\right),x_{f}^{\prime }\left(j,2\right),\ldots ,x_{f}^{\prime }\left(j,T\right)\right]}^{\prime }$. Across all *J* sensors and *T* time instants, the $TG_{f}\times J$ matrix of complex amplitudes is defined as $X_{f}=\left[x_{f}^{*}\left(1\right),x_{f}^{*}\left(2\right),\ldots ,x_{f}^{*}\left(J\right)\right]$. The solution to the problem being formulated relies on the assumption that vector x(j,t) has sparsity. Sparsity entails vector x(j,t) having non-zero entries corresponding only to the active frequencies at each time instant *t*.

A sparse estimate of the frequency hopping pattern per source $X_{f}$ (or equivalently an estimate of $x_{f}\left(j,t\right)$ for all *j* and all *t*) is denoted as ${\hat{X}}_{f}$ (or ${\hat{x}}_{f}\left(j,t\right)$), and it is found by solving the following optimization problem:(4)$${\hat{X}}_{f}=\underset{x_{f}}{\arg\;\min}{∥Y^{\prime }-H_{f}X_{f}∥}_{2}^{2}+\lambda_{f}\sum_{m=1}^{M}\sum_{t=1}^{T}{∥x_{f}\left(j,t\right)∥}_{1}.$$

The $\ell_{1}$-norm in (4) enforces the sparsity constraint. Hperparameter $\lambda_{f}$ controls the sparsity of the solution. A large $\lambda_{f}$ increases the penalty of non-zero elements of $x_{f}\left(j,t\right)$.

### 3.2. DOA Estimation

To formulate the problem of estimating the DOA of sources at each time instant 0≤t≤T as a sparse representation problem, we again define a dictionary matrix and a matrix of unknowns. A grid comprising $G_{d}$ possible DOAs, $\Theta =[\theta_{1},\theta_{2},\ldots ,\theta_{G_{d}}]$, is used to define the $J\times G_{d}$ DOA modal dictionary
(5)$$H_{d}=\left[a\left(\theta_{1}\right),a\left(\theta_{2}\right),\ldots ,a\left(\theta_{G_{d}}\right)\right],$$
where $a(\theta_{i})$ is the steering vector associated with DOA $\theta_{i}$ defined as $a\left(\theta_{i}\right)={\left[1,a\left(\theta_{i}\right),\ldots ,a^{J-1}\left(\theta_{i}\right)\right]}^{\prime }$ (see Equation (Equation 1) for other definitions).

We let $x_{d}\left(t\right)$ be the $G_{d}$-length vector of unknown complex amplitudes associated with grid DOAs at time *t*. Across all *T* sampling instants, the $G_{d}\times T$ matrix of complex amplitudes is defined as $X_{d}=\left[x_{d}\left(1\right),x_{d}\left(2\right),\ldots ,x_{d}\left(T\right)\right]$. Given observations Y, a sparse estimate of the DOA pattern $X_{d}$ (or, equivalently, an estimate of $x_{d}\left(t\right)$ for all (*t*) is denoted as ${\hat{X}}_{d}$ (or ${\hat{x}}_{d}\left(t\right)$), and it is found by solving the following optimization problem:(6)$${\hat{X}}_{d}=\underset{x_{d}}{\arg\;\min}{∥Y-H_{d}X_{d}∥}_{2}^{2}+\lambda_{d}\sum_{t=1}^{T}{∥x_{d}\left(t\right)∥}_{1}.$$

Similar to the optimization problem for estimating the frequency hops, the formulation includes hyperparameter $\lambda_{d}$ that controls sparsity.

### 3.3. Hidden State Filtering

We let ${\hat{x}}_{d}\left(t\right)$ denote a non-zero component of solution ${\hat{x}}_{d}\left(t\right)$ to the optimization problem in Equation (Equation 6). Then, ${\hat{x}}_{d}\left(t\right)$ is associated with a single source, and if state s(t) of the source were known, ${\tilde{x}}_{d}\left(t\right)={\hat{x}}_{d}\left(t\right)\cdot s\left(t\right)$ would be a refined estimate of ${\hat{x}}_{d}\left(t\right)$ in the sense that a spurious component would be removed from the solution to Equation (Equation 6) if ${\hat{x}}_{d}\left(t\right)\neq 0$ but s(t)=0. Conversely, the solution would stand if s(t)=1. While state s(t) is hidden to the observer, it may be inferred from state observations. We let set $C=\left\{x|x\neq 0\right\}$; then, observation z(t) of the state of a source is obtained as $z\left(t\right)=1_{C}\left\{{\hat{x}}_{d}\left(t\right)\right\}$, where $1_{C}$ denotes the indicator function of set *C*. Hidden state filtering is the problem of inferring state sequence s(1:T) for each source given an observation sequence z(1:T) of the source and an HMM model Ω. In the remainder of this subsection, we discuss the forward–backward procedure, which is then used to solve the HSF problem. The algorithms implementing these methods assume that the HMM parameter Ω is known. The estimation of parameter Ω, when it is not known *a priori*, is addressed in the subsequent subsection.

#### 3.3.1. Forward–Backward Procedure

The forward–backward procedure comprises the iterative calculation of forward and backward variables given the observed sequence and model parameter Ω. Forward variable $\alpha_{j}\left(t\right)$ is defined as the probability of a state at time *t* given the observed sequence up to and including time *t*:(7)$$\alpha_{j}\left(t\right)≜P\left(s\left(t\right)=j|z\left(1:t\right),\Omega \right),j=0,1.$$

Solving for $\alpha_{j}\left(t\right)$ consists of the following steps [41]:1.Initialization:
(8)$$\alpha_{i}\left(1\right)=\pi_{i}b_{i}\left(z\left(1\right)\right),i=0,1.$$2.Induction:
(9)$$\begin{array}{c} \alpha_{j}\left(t+1\right)=\left[\sum_{i\in \{0,1\}}\alpha_{i}\left(t\right)a_{ij}\right]b_{j}\left(z\left(t\right)\right),\\ j=0,1,0\leq t\leq T-1. \end{array}$$3.Termination:
(10)$$P\left(z\left(1:T\right)|\Omega \right)=\sum_{i=0,1}\alpha_{i}\left(T\right).$$

Backward variable $\beta_{i}\left(t\right)$ is defined as the probability of the observation sequence from t+1 to the end of the sequence, conditioned on state s(t)=i
(11)$$\beta_{i}\left(t\right)≜P\left(z\left(t+1:T\right)|s\left(t\right)=i,\Omega \right),i=0,1.$$

Solving for $\beta_{i}\left(t\right)$ consists of the following steps [41]:1.Initialization:
(12)$$\beta_{i}\left(T\right)=1,\;i=0,1.$$2.Induction
(13)$$\begin{array}{c} \beta_{i}\left(t\right)=\sum_{j\in \{0,1\}}a_{ij}b_{j}\left(z\left(t+1\right)\right)\beta_{j}\left(t+1\right),\\ i=0,1,t=T-1,T-2,\ldots ,1. \end{array}$$
HSF methods discussed next use the forward and backward variables.

#### 3.3.2. Individually Most Probable States

There are different ways of finding the “optimal” state sequence s(1:T) given the observed sequence z(1:T) and the HMM parameter Ω. One reasonable criterion is to choose, at each time *t*, the state that is most probable given the observed sequence and the HMM parameter. We denote the belief state as follows:(14)$$\gamma_{i}\left(t\right)≜P\left(s\left(t\right)=i|z\left(1:T\right),\Omega \right).$$
It can be shown that variable $\gamma_{i}\left(t\right)$ may be expressed in terms of the forward–backward variables [41]:(15)$$\gamma_{i}\left(t\right)=\frac{\alpha_{i}\left(t\right)\beta_{i}\left(t\right)}{\sum_{j\in \left\{0,1\right\}}\alpha_{j}\left(t\right)\beta_{j}\left(t\right)}.$$
The most probable estimate (MPE) to a state at time *t* is the solution to the following problem:(16)$${\hat{s}}_{\mathrm{MPE}}\left(t\right)=\underset{i=0,1}{\arg\max}\;\gamma_{i}\left(t\right).$$

We note that the MPE to a state may be computed directly from the forward–backward variables. The MPE state sequence for a source is expressed as follows:(17)$${\hat{s}}_{\mathrm{MPE}}\left(1:T\right)=\left\{{\hat{s}}_{\mathrm{MPE}}\left(1\right),{\hat{s}}_{\mathrm{MPE}}\left(2\right),\ldots ,{\hat{s}}_{\mathrm{MPE}}\left(T\right)\right\}.$$
Using the MPE state sequence, the matrix of complex amplitudes containing the filtered vectors can be calculated as outlined in Algorithm 1.
**Algorithm 1** Hidden State Filtering (MPE)1:**Input:** Matrix of complex amplitudes ${\hat{X}}_{d}$ and HMM model paramter Ω2:**Output:** Inferred state sequence ${\hat{s}}_{\mathrm{MPE}}\left(1:T\right)$3:**for** each non-zero row of ${\hat{X}}_{d}$ **do**4:   **for** t=1:T **do**5:     $z\left(t\right)=1_{c}\left\{{\hat{x}}_{d}\left(t\right)\right\}$6:     Calculate ${\hat{s}}_{\mathrm{MPE}}\left(t\right)$ from Equation (Equation 16)7:   **end for**8:**end for**

#### 3.3.3. Most Probable Sequence of States

The most probable sequence of states given the observed sequence of a source is known as the *maximum* a posteriori (MAP) estimate and is given by
(18)$${\hat{s}}_{\mathrm{MAP}}\left(1:T\right)=\underset{s(1:T)}{\arg\max}P\left(s\left(1:T\right)|z\left(1:T\right)\right).$$
The MAP estimate may be computed using the well-known Viterbi algorithm [42]. The following quantity is defined:(19)$$\rho_{i}\left(t\right)≜\max_{s(1:t-1)}\;P\left(s\left(1:t-1\right),s\left(t\right)=i|z\left(1:t\right),\Omega \right),$$
which represents the joint probability of reaching state *i* at time *t* and taking the most probable path. A recursive expression for this probability is obtained noting that if $\rho_{i}\left(t-1\right)$ is known, then $\rho_{j}\left(t\right)$ may be computed by accounting for the transition from state *i* at time t−1 to state *j* at time *t*. This probability may be computed using quantities previously defined according to
(20)$$\rho_{j}\left(t\right)=\left(\max_{i}\;\rho_{i}\left(t-1\right)\right)a_{ij})b_{j}\left(z\left(t\right)\right).$$
The argument that maximizes Equation (Equation 20) is denoted as $\psi_{j}\left(t\right)≜\underset{i=0,1}{\arg\max}\;\left(\rho_{i}\left(t-1\right)a_{ij}\right)$. Following this procedure, the first state that can be determined as part of the most probable path is the final state s(T), since any earlier determination may be affected by later times. The algorithm is summarized as follows:1.Initialization:
(21a)$$\rho_{i}\left(1\right)=\pi_{i}b_{i}\left(z\left(1\right)\right),\;i=0,1,$$
(21b)$$\psi_{i}\left(1\right)=0,\;i=0,1.$$2.Recursion:
(22a)$$\rho_{j}\left(t\right)=\left(\max_{i}\;\rho_{i}\left(t-1\right)a_{ij}\right)b_{j}\left(z\left(t\right)\right),\;2\leq t\leq T,\;j=0,1,$$
(22b)$$\psi_{j}\left(t\right)≜\underset{i=0,1}{\arg\max}\;\left(\rho_{i}\left(t-1\right)a_{ij}\right),\;2\leq t\leq T,\;j=0,1.$$3.Termination:
(23)$${\hat{s}}_{\mathrm{MAP}}\left(T\right)=\underset{i=0,1}{\arg\max}\;\left(\rho_{i}\left(T\right)\right).$$4.MAP path:
(24)$${\hat{s}}_{\mathrm{MAP}}\left(t\right)=\psi_{\hat{s}\left(t+1\right)}\left(t+1\right),\;t=T-1,T-2,\ldots ,1.$$

Using the MAP path, the matrix of complex amplitudes containing the filtered vectors can be calculated as outlined in Algorithm 2.
**Algorithm 2** Hidden State Filtering (MAP)1:**Input:** Matrix of complex amplitudes ${\hat{X}}_{d}$ and HMM model parameter Ω2:**Output:** Inferred state sequence ${\hat{s}}_{\mathrm{MAP}}\left(1:T\right)$3:**for** each non-zero row of ${\hat{X}}_{d}$ **do**4:   **for** t=1:T **do**5:     $z\left(t\right)=1_{c}\left\{{\hat{x}}_{d}\left(t\right)\right\}$6:     Calculate ${\hat{s}}_{\mathrm{MAP}}\left(t\right)$ from Equation (Equation 24)7:   **end for**8:**end for**

#### 3.3.4. Learning HMM Parameters

The HSF methods discussed previously assume the HMM parameters Ω=(A,B,π) are known. In practice, however, it cannot be assumed that the model is known in a blind scenario. The goal here is to describe a method by which Ω is estimated based on the observed sequences. Here, the observed sequence refers to the observed activity of a source. The HMM parameters are learnt from the observed sequences of the sources detected as active according to $\hat{\Omega }=\underset{\Omega }{\arg\max}\;P(z\left(1:T|\Omega \right))$. For the first active source, the HMM parameter Ω=(A,B,π) is randomly initialized. For the subsequent active sources, Ω obtained from the previous active source is used for initialization. The parameter obtained from the last active source is then used to calculate the hidden state sequences using the MPE and MAP methods. For the purpose of estimating the HMM parameter Ω, the following quantity is defined:(25)$$\xi_{i,j}\left(t\right)=P\left(s\left(t\right)=i,s\left(t+1\right)=j|z\left(1:T\right),\Omega \right).$$

It can be shown that variable $\xi_{i,j}\left(t\right)$ may be expressed in terms of forward–backward variables [41]:(26)$$\xi_{i,j}\left(t\right)=\frac{\alpha_{i}\left(t\right)a_{ij}b_{j}\left(z\left(t+1\right)\right)\beta_{j}\left(t+1\right)}{\sum_{i=0,1}\sum_{j=0,1}\alpha_{i}\left(t\right)a_{ij}b_{j}\left(z\left(t+1\right)\right)\beta_{j}\left(t+1\right)}.$$

Utilizing the forward variable $\alpha_{j}\left(t\right)$ defined in Equation (Equation 7), the backward variable $\beta_{i}\left(t\right)$ defined in Equation (Equation 11), the belief state $\gamma_{i}\left(t\right)$ defined in Equation (Equation 14) and the quantity $\xi_{i,j}\left(t\right)$ defined in Equation (Equation 25), the HMM parameters are learned in the following steps:1.Initialization: Randomly initialize Ω=(A,B,π) for the first active source.2.For an estimated active source:(a)Use Equations (Equation 8)–(Equation 10) to calculate $\alpha_{j}\left(t\right)$ and Equations (Equation 12) and (Equation 13) to calculate $\beta_{i}\left(t\right)$.(b)Use Equation (Equation 15) to calculate $\gamma_{i}\left(t\right)$ and Equation (Equation 26) to calculate $\xi_{i,j}\left(t\right)$.(c)Update model parameters:
(27a)$${\hat{\pi }}_{i}=\gamma_{i}\left(1\right),$$
(27b)$${\hat{a}}_{ij}=\frac{\sum_{t=1}^{T-1}\xi_{i,j}\left(t\right)}{\sum_{t=1}^{T-1}\gamma_{i}\left(t\right)},$$
(27c)$${\hat{b}}_{j}\left(k\right)=\frac{\sum_{t=1}^{T}1\left(z\left(t\right)=k\right)\gamma_{j}\left(t\right)}{\sum_{t=1}^{T}\gamma_{j}\left(t\right)}.$$Here, 1(a) is the indicator function, i.e., 1(a)=1 if *a* is true, and 0 otherwise.(d)Set $\hat{\Omega }=\left(\hat{A},\hat{B},\hat{\pi }\right)$, and repeat step (2).
The last HMM parameters $\hat{\Omega }=\left(\hat{A},\hat{B},\hat{\pi }\right)$ obtained after running through the observed sequences of all sources is considered as the common model for all sources. The HMM parameter learning is outlined in Algorithm 3.
**Algorithm 3** Learning HMM parameters1:**Input:** Matrix of complex amplitudes ${\hat{X}}_{d}$2:**Output:** Learnt HMM parameters $\hat{\Omega }=\left(\hat{A},\hat{B},\hat{\pi }\right)$3:Initialize Ω=(A,B,π).4:**for** each non-zero row of ${\hat{X}}_{d}$ **do**5:   **for** T=1:t **do**6:     $z\left(t\right)=1_{c}\left\{{\hat{x}}_{d}\left(t\right)\right\}$7:     **while** no convergence **do**8:        Obtain $\alpha_{j}\left(t\right)$ and $\beta_{i}\left(t\right)$ using Forward-Backward procedure9:        Update $\hat{\Omega }=\left(\hat{A},\hat{B},\hat{\pi }\right)$ using Equations ([Disp-formula FD27a-entropy-25-01292])–([Disp-formula FD27c-entropy-25-01292])10:     **end while**11:     $\Omega \leftarrow \hat{\Omega }$.12:   **end for**13:**end for**

### 3.4. Pairing

Given the FH and DOA estimates, the final stage is to pair them. For each time instant *t*, the pairing stage is designed to pick a combination of DOA and FH estimates that provide the best fit to the observed data.

To perform the pairing, two new dictionaries ${\tilde{H}}_{d}$ and ${\tilde{H}}_{f}$ are formed from the original respective dictionaries $H_{d}$ and $H_{f}$. The new dictionary ${\tilde{H}}_{d}$ is defined as a submatrix of $H_{d}$, with elements corresponding to non-zero entries in ${\tilde{X}}_{d}$. Similarly, the new dictionary ${\tilde{H}}_{f}$ is defined as a submatrix of $H_{f}$, with elements corresponding to non-zero entries in ${\hat{X}}_{f}$. The new dictionaries are introduced to limit the computational cost of the pairing operation and used to create a new dictionary [43]. The Kronecker product of ${\tilde{H}}_{d}$ and ${\tilde{H}}_{f}$ defines this new dictionary $\tilde{H}$:(28)$$\tilde{H}={\tilde{H}}_{d}⊗{\tilde{H}}_{f}.$$

This dictionary is a grid that contains all active frequencies for each active source over the entire observation interval. The pairing stage utilizes the newly formed dictionary $\tilde{H}$ to output matrix X, which contains the complex amplitudes of sources which are indexed by their hop frequencies and their DOAs. The following optimization problem is solved:(29)$$\hat{X}=\underset{x}{\arg\;\min}{∥Y-\tilde{H}X∥}_{2}^{2}+\lambda \sum_{t=1}^{T}{∥x(t)∥}_{1}$$
to obtain a sparse vector x(t) whose non-zero elements are the estimated complex amplitudes. Here, λ is the hyperparameter that controls the sparsity of x(t). The pairing of FH estimates and source activity is outlined in Algorithm 4.
**Algorithm 4** Pairing of source activity and FH estimates1:**Input:** Matrices ${\tilde{X}}_{d}$, ${\hat{X}}_{f}$ and dictionaries $H_{d}$, $H_{f}$2:**Output:** Matrix of paired source activity and FH estimates $\hat{X}$3:Obtain new dictionary ${\tilde{H}}_{d}$ from $H_{d}$ using $X_{d}$4:Obtain new dictionary ${\tilde{H}}_{f}$ from $H_{f}$ using $X_{f}$5:Create dictionary $\tilde{H}$ using Equation (Equation 28)6:Obtain $\hat{X}$ from Equation (Equation 29)

Source separation is obtained by labeling the sources according to DOA estimates. A source label may be associated with multiple frequency hops. Furthermore, the pairing stage is also capable of reducing false alarms in the FH and DOA estimation stages by considering a source to be active at a given time instant only if it produces a joint FH and DOA estimate.

## 4. Results

In this section, numerical results are presented to demonstrate the performance of the proposed approach to separate intermittent FH signals. We assume that the activity for each source is defined by an HMM with transition probabilities $a_{01}=0.02$ and $a_{10}=0.02$. A modal dictionary for DOA estimation is considered with $G_{d}=36$ bins, each having a bin size of 5 degrees. The FH grid has $G_{f}=40$ bins with a spacing of 50 kHz for a 2 MHz total bandwidth. The signals being transmitted are slow FH signals in which each hop contains one or more symbols. Frequency hops do not violate the narrowband assumption for DOA estimation. The number of unresolveable multipaths per cluster is M=20 and the deviation from mean DOA of each cluster has a variance of $\sigma_{\theta }^{2}=2$ degrees. In the following figures, we consider two clusters per source while referring to sources in the multiple-cluster model. The weights of the various optimization problems included in this paper are hyperparameters, and they are chosen in accordance with methods expanded in [37]. Experiments are run on Matlab R2020a on a computer with Intel^®^ Core^TM^ i5 CPU and Windows OS.

The performance criterion chosen is the receiver operating characteristic (ROC), in which the probability of correct detection $P_{d}$ is plotted against the probability of false alarm $P_{fa}$. The probability of correct detection for source activity is computed as the ratio of the number of correctly detected sources to the number of true active sources. A source activity is deemed a correct detection if it has a “similar” source activity over the total observation interval to a true active source. Multiple activity patterns associated with different DOAs are considered “similar” if they match over a prescribed fraction of the time samples at which their values are one. Through experimentation, this fraction is chosen to be 0.95. The probability of false alarm for source activity is the ratio of the number of spuriously detected sources over the entire observation interval to the number of true active sources. Source activity is spurious if a source is detected to be active when no true source is active. Analogous definitions apply to $P_{d}$ and $P_{fa}$ of frequency hops and to paired activity.

Figure 5 shows the ROC of FH estimation for 5 sources with an SNR per source of 10 dB. For $P_{fa}=0.3$, FH estimation $P_{d}=0.98$. The performance of the FH estimation affects the source separation, as the pairing uses the estimations from both the previous stages to pick a pair of source activity and FH pattern that best fits the received signals at each time instant. This is demonstrated later, in the ROC of paired activity.

Figure 6 demonstrates the effect of HSF on the correct detection of activity for 5 LOS sources with an SNR per source of 10 dB. The figure shows the ROC performance with two HSF techniques. The ROC is obtained by solving Equation (Equation 6), followed by the thresholding of ${\hat{X}}_{d}$. The threshold ranges from the smallest to the largest values in matrix ${\hat{X}}_{d}$ obtained by solving Equation (Equation 6). Each point of the ROC, consisting of a pair of probability of correct detection and probability of false alarm, corresponds to one of these threshold values. Without filtering, for $P_{fa}$ of activity =0.3, $P_{d}$ of activity =0.67. For the same $P_{fa}$, with MPE applied, $P_{d}=0.86$ when Ω is learned, and $P_{d}=0.87$ when Ω is known. With MAP applied, $P_{d}=0.96$ when Ω is learned, and $P_{d}=0.97$ when Ω is known.

Figure 7 and Figure 8 demonstrate the effect of HSF on the correct detection of activity for single-cluster and multiple-cluster propagation environments, respectively. DOA estimates in these figures are considered “similar” when the fraction measuring similarity is 0.95. Improvement in performance is observed in activity when HSF is applied. In the next figures, we apply MAP filtering with unknown HMM parameter Ω only.

The performance of pairing is demonstrated next. Figure 9 shows the ROCs of paired activity for all propagation environments, LOS, single cluster and multiple cluster, when the fraction measuring similarity is 0.95. For $P_{fa}=0.3$, $P_{d}=0.94$ for the LOS propagation environment, $P_{d}=0.84$ for the single-cluster propagation environment, and $P_{d}=0.83$ for the multiple-cluster propagation environment.

Figure 10 plots the probability of detection as a function of the number of LOS intermittent sources that are detected as active during the observation interval. The detection threshold is set at a probability of false alarm $P_{fa}=0.15$. It is observed that the performance of both the FH and DOA estimation stages affect the performance of the pairing stage.

## 5. Discussion

In this paper, an approach is proposed to solve the BSS problem for FH sources that are stationary, spatially sparse and have intermittent activity observed over channels that follow the spatial channel model for three propagation environments. The source memory is utilized in the filtering of source activity to enhance performance. Current methods in the literature do not perform the association of multiple frequency hops to the sources they are transmitted from. We bridge this gap by pairing the FH estimates with DOA estimates and labeling signals to their sources, irrespective of their hopped frequencies. The pairing stage uses the estimations from the previous stages to pick a pair of the DOA and FH patterns that best fits the received signals at each time instant. The pairing assigns source labels to the signals and is capable of reducing false alarms that arise in the individual stages, thus improving the accuracy of the proposed approach as confirmed by numerical results. Numerical results demonstrate that HSF improves BSS performance, and that MAP HSF outperforms MPE HSF.

## Figures and Tables

**Figure 1 entropy-25-01292-f001:**
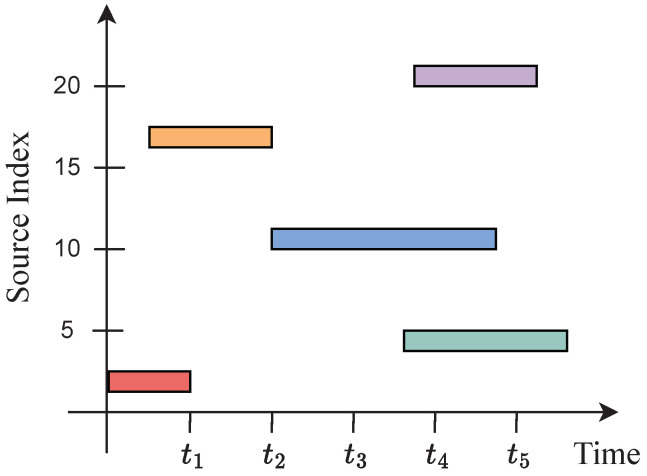
Intermittent source activity. The filled blocks denote active sources at different time instants.

**Figure 2 entropy-25-01292-f002:**
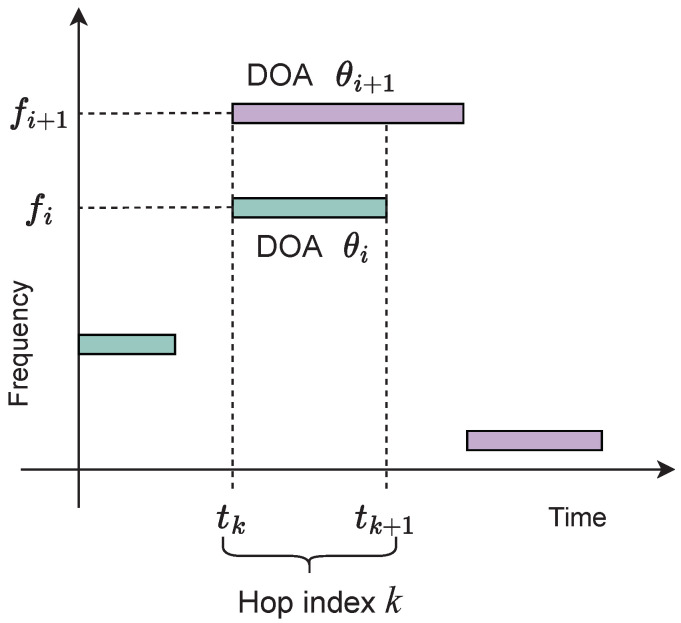
Source activity of two FH sources depicted by filled blocks with distinct shading. Time duration $t_{k}\leq t<t_{k+1}$ denotes a single hop with hop index *k* where sources with mean DOAs $\theta_{i}$ and $\theta_{i+1}$ transmit with frequencies $f_{i}$ and $f_{i+1}$, respectively.

**Figure 3 entropy-25-01292-f003:**
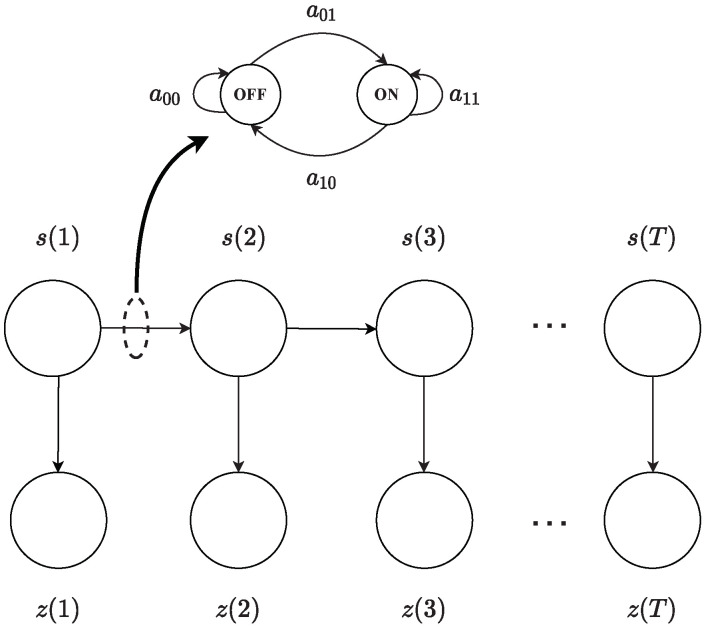
Hidden Markov model (HMM) for a source.

**Figure 4 entropy-25-01292-f004:**
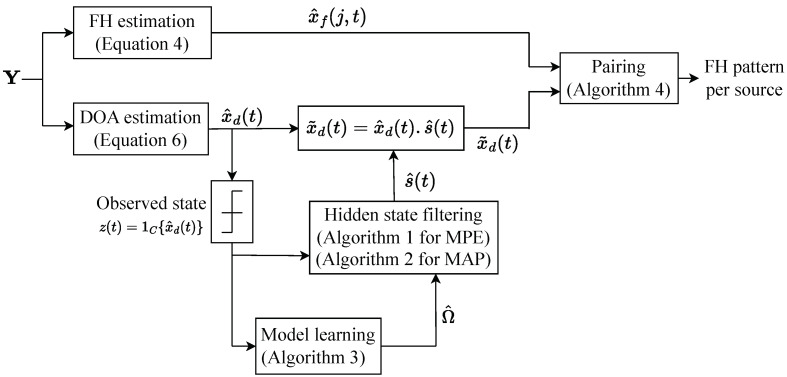
Block diagram of the proposed algorithm to separate multiple FH sources.

**Figure 5 entropy-25-01292-f005:**
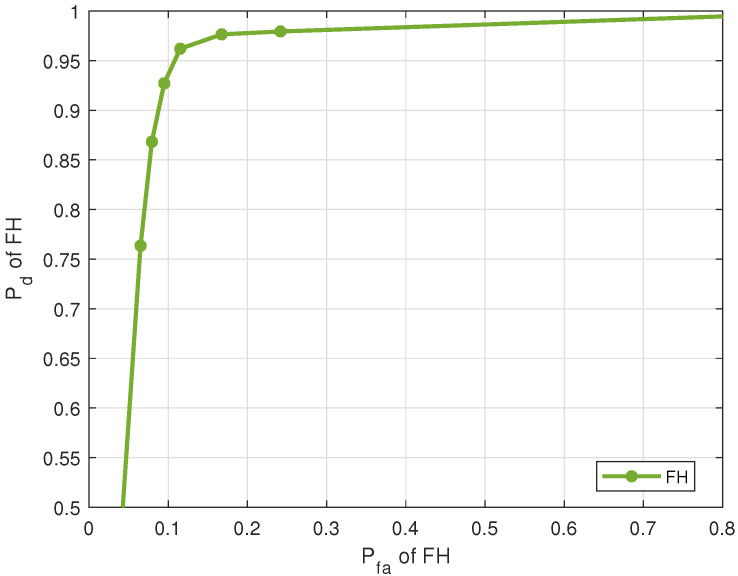
$P_{d}$ versus $P_{fa}$ of FH estimates (5 sources, J=20 sensors, T=1000 samples, SNR = 10 dB).

**Figure 6 entropy-25-01292-f006:**
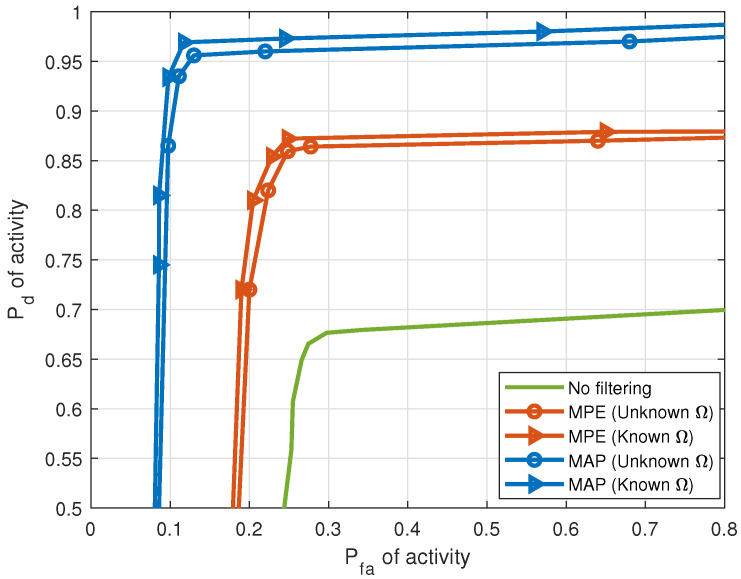
$P_{d}$ versus $P_{fa}$ of activity with and without HSF for LOS sources (5 sources, J=20 sensors, T=1000 samples, SNR = 10 dB).

**Figure 7 entropy-25-01292-f007:**
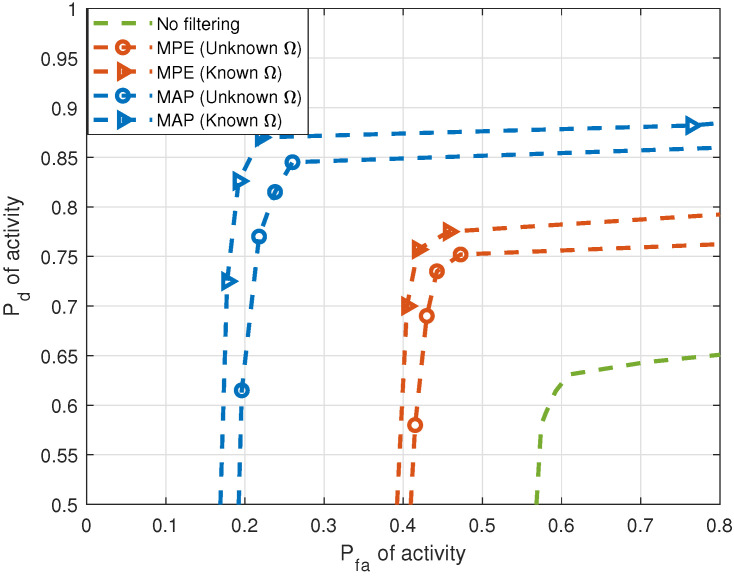
$P_{d}$ versus $P_{fa}$ of activity with and without HSF for single-cluster propagation environment (5 sources, J=20 sensors, T=1000 samples, SNR = 10 dB).

**Figure 8 entropy-25-01292-f008:**
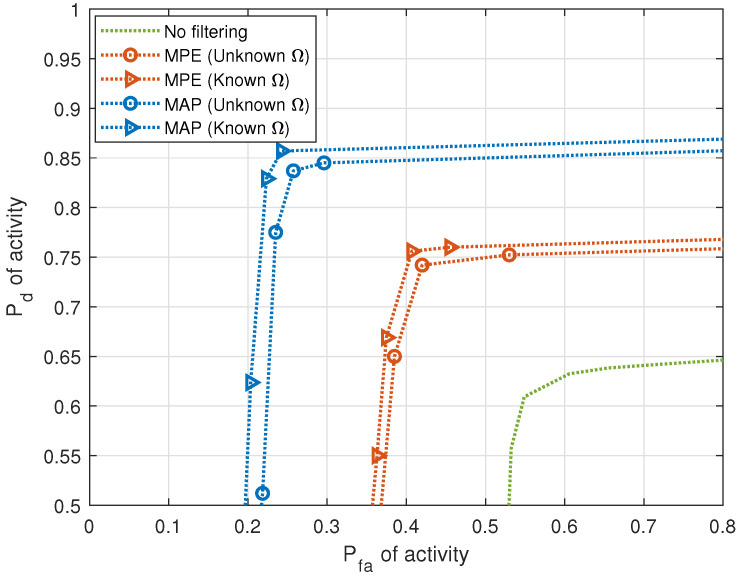
$P_{d}$ versus $P_{fa}$ of activity with and without HSF for multiple-cluster propagation environment (5 sources, J=20 sensors, T=1000 samples, SNR = 10 dB).

**Figure 9 entropy-25-01292-f009:**
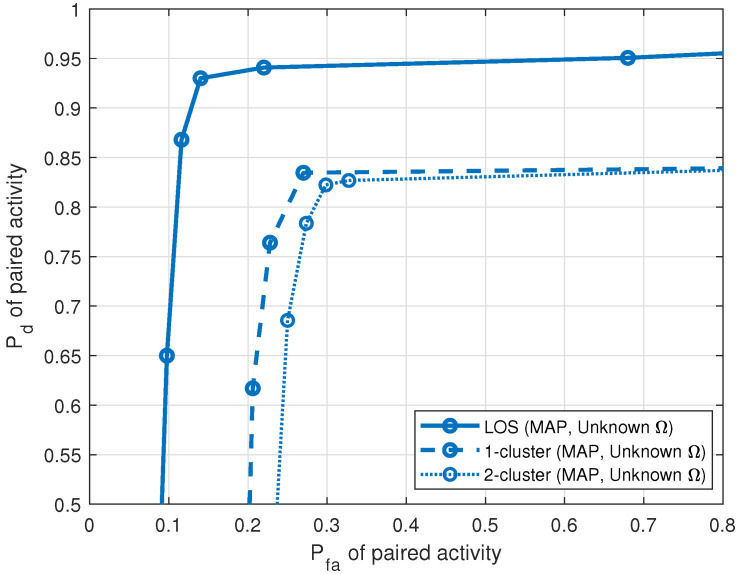
$P_{d}$ versus $P_{fa}$ of paired activity for LOS, single-cluster and multiple-cluster propagation environments (5 sources, J=20 sensors, T=1000 samples, SNR = 10 dB).

**Figure 10 entropy-25-01292-f010:**
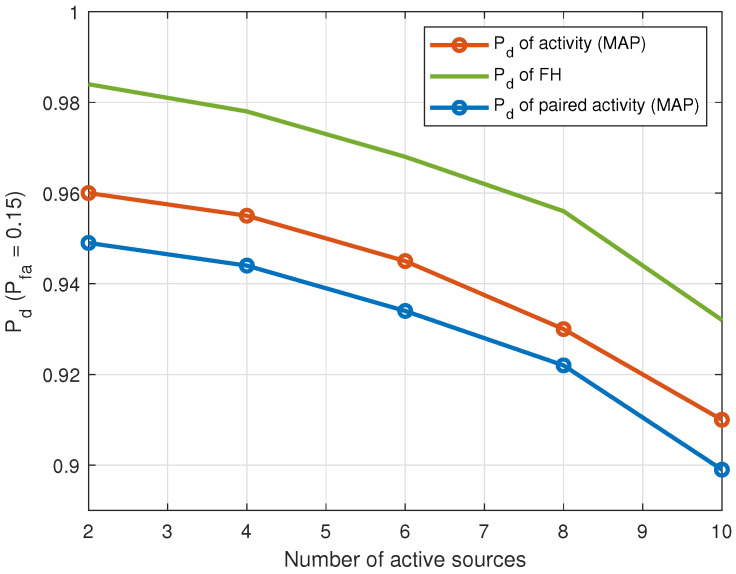
$P_{d}$ when $P_{fa}=0.15$ versus number of intermittent sources detected as active during observation interval in the LOS propagation environment (J=20 sensors, T=1000 samples, SNR = 10 dB).

**Table 1 entropy-25-01292-t001:** Short descriptions and limitations of approaches from current literature.

References	Description of Approach	Limitations
[20,21,22,23,30]	Time-frequency analysis to obtain representation of FH signals	Not possible to obtain good resolution in both time and frequency domains; cross-term interference; spectral leakage; high SNR requirements
[26]	TFA as an exploratory tool, applying particle filter on initial estimations	Limited by the assumption that only one FH signal is present
[27]	Blind ML-based iterative algorithm to estimate hop timing and frequency	Cannot be generalized to multiple FH signals
[28]	Dynamic programming ML estimator for multiple signals	Treats each hop as a distinct source; cannot be used for multiple sources that each transmit at multiple frequencies by hopping
[29]	Sparse linear regression for multiple FH signals	
[31,32,33]	Joint estimation of FH parameters and DOA	Assumes all sources are active for the entire observation period
[34]		Assumes all hop periods are uniform

## Data Availability

The data presented in this study are available in this article and the article titled “Interference Mitigation in Blind Source Separation by Hidden State Filtering”, in proceedings of the 57th Annual Conference on Information Sciences and Systems (CISS), Baltimore, MD, USA, 22–24 March 2023.

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
