# Peer review of "Blind Source Separation of Intermittent Frequency Hopping Sources over LOS and NLOS Channels†"

_entropy, 2023, doi:10.3390/e25091292_

Round 1

Reviewer 1 Report

Blind source separation method over LOS and NLOS channels is presented  in this manuscript. Current version contains several major concerns.

1. What is the main innovation and advantages of this work? The authors should summarize these two points carefully, reasonably and convincingly in the Abstract, Introduction and try to highlighed them in Conclusions.

2. What is the main difference as the proposed method applied in LOS and NLOS channels? If there is no difference, how to ensure it?

3. A comparison table with other similar works should be included to show the main selling-point of the proposed method.

4. In Section 4 Results, the computation source for the simulations should be included, so that the readers can evaluate the main efficiency of the proposed method.

Reviewer 2 Report

see joint PDF document.

Round 2

Reviewer 1 Report

no more comments

Reviewer 2 Report

The previous issues have been all addressed by the authors.

I think that this version is appropriate for publication.